# De novo compartment deconvolution and weight estimation of tumor samples using DECODER

Xianlu Laura Peng [1], Richard A. Moffitt [2], Robert J. Torphy [3], Keith E. Volmar[4] & Jen Jen Yeh [1,5,6]*

Tumors are mixtures of different compartments. While global gene expression analysis profiles the average expression of all compartments in a sample, identifying the specific contribution of each compartment remains a challenge. With the increasing recognition of the importance of non-neoplastic components, the ability to breakdown the gene expression contribution of each is critical. Here, we develop DECODER, an integrated framework which performs de novo deconvolution and single-sample compartment weight estimation. We use DECODER to deconvolve 33 TCGA tumor RNA-seq data sets and show that it may be applied to other data types including ATAC-seq. We demonstrate that it can be utilized to reproducibly estimate cellular compartment weights in pancreatic cancer that are clinically meaningful. Application of DECODER across cancer types advances the capability of identifying cellular compartments in an unknown sample and may have implications for identifying the tumor of origin for cancers of unknown primary.

[1] Lineberger Comprehensive Cancer Center, University of North Carolina, Chapel Hill, NC, USA. [2] Department of Biomedical Informatics, Stony Brook University, Stony Brook, NY, USA. [3] Department of Surgery, University of Colorado, Denver, CO, USA. [4] University of North Carolina–Rex Healthcare, Chapel Hill, NC, USA. [5] Departments of Surgery, University of North Carolina, Chapel Hill, NC, USA. [6] Department of Pharmacology, University of North Carolina, Chapel Hill, NC, USA. *email: jen_jen_yeh@med.unc.edu

Tumor samples are mixtures of distinct cell populations that contribute to intra-tumor heterogeneity, including immune, stroma, and normal cells[1,2]. Therefore, with bulk tumor samples, the analysis of tumor gene expression can be significantly confounded by the presence of nonneoplastic cell types, while the contribution of the tumor microenvironment is difficult to separate. Although laser-capture microdissection (LCM) and single-cell sequencing techniques strive to tackle these problems, both of them present certain limitations. LCM is labor intensive and may influence the quality of the microdissected tissue for further analysis[3,4]. Single-cell sequencing is still expensive, computing resource heavy, and currently limited by the lack of comprehensive cell-sorting biomarkers[5,6].

To eliminate the need of relying on LCM or single-cell-based techniques, a plethora of computational strategies have been developed to deconvolve the mixed signal present in a bulk tumor sample using RNA gene expression, DNA copy number data, or DNA methylation data. Algorithms based on DNA copy-number alterations, e.g., ABSOLUTE[7], and DNA methylation profiles, e.g., MethylPurify[8] and InfiniumPurify[9], focus on inferring tumor purity, while expression-based deconvolution methods mainly handle estimation of compartment fractions, as well as extraction of compartment-specific expression profiles[2]. However, the current expression-based deconvolution methods still pose a number of limitations. Some methods are limited to the presupposition of a certain combination of compartments, such as DeMix (tumor and normal)[10], UNDO (tumor and stroma)[11], and ESTIMATE (tumor, stroma and immune)[12]. Other methods, such as DeconRNAseq[13] and CIBERSORT[14], provide the flexibility to measure any number of specific compartments. However, they require knowledge of the pure expression of compartments as the reference, which is difficult to obtain in practice. Similarly, DSA[15] requires lists of marker genes. However, frequently the exact compartments in a sample are unknown, and samples are inherently heterogenous. Therefore, the incomplete knowledge of the number of compartments may hamper the accuracy of calculations of the compartment proportions. In addition, like many other quadratic programming-based algorithms, DSA[15] has a minimum sample size requirement to perform the estimation, imposing a need for larger data sets.

Pancreatic ductal adenocarcinoma (PDAC) is characterized by relatively low tumor purity and large amounts of desmoplastic stroma. Therefore, identifying tumor-specific alterations in PDAC is a continuing challenge. To perform virtual microdissection and study compartment-specific signatures, we previously deconvolved bulk PDAC samples by adapting the nonnegative matrix factorization (NMF) algorithm[16]. As a result, we identified two tumor-specific (Basal-like and Classical) and two stroma-specific (Activated and Normal) subtypes, together with exocrine, endocrine, and immune factors. Like other standard NMF applications, the number of factors ($K$) that the input matrix may be decomposed into must be determined a priority. Although the performance of NMF at different $K$ may be evaluated by silhouette and cophenetic correlation coefficient, this evaluation assumes the exclusive classification of each sample into one of the $K$ clusters[17,18], which may not be as biologically clear cut. In our previous study, we empirically determined $K$ by dedicated manual association of biological relevance to each factor at multiple trial runs of $K$, which can be time consuming and resource intensive[16]. Thus, developing a streamlined framework that is able to automatically determine $K$ is very appealing and will have potential applicability to the bulk tumor sample deconvolution of any cancer type.

Here, we present de novo compartment deconvolution and weight estimation of tumor samples (DECODER), an NMF-based integrated and sophisticated framework for the de novo deconvolution of tumor mixture samples for compartments, and estimation of compartment weights for samples (Fig. 1a). We show that DECODER can perform automated de novo deconvolution for patient samples to derive dynamic and biologically sound compartments without setting ad hoc parameters. By applying DECODER to RNA-sequencing (RNA-seq) data from 33 The Cancer Genome Atlas (TCGA) cancer types, 269 cancer-specific compartments are identified, with a list of marker genes for each compartment. In addition, DECODER can be used to calculate the compartment weights for a single sample, making it potentially applicable in the clinical setting. By applying DECODER to PDAC microarray[16], TCGA pancreatic adenocarcinoma (PAAD)[19], COMPASS[20], and ICGC (International Cancer Genome Consortium) pancreatic cancer[18] RNA-seq data sets, we demonstrate that DECODER provides insight into pancreatic cancer biology with clinical implications. This framework is not only a useful algorithm for de novo and single-sample deconvolution of heterogenous samples, but also to the best of our knowledge, for the first time, provides cancer type-specific derived compartment information.

## Results

**PDAC compartments faithfully captured in microarray data set**. Given a bulk measurement **A**, which is a $N \times M$ matrix of $N$ rows of genes and $M$ columns of samples, the aim of the de novo deconvolution is to factorize **A** into two matrices **W** and **H**, where **W** is a $N \times K$ matrix, **H** is a $K \times M$ matrix, and $K$ is the number of compartments (Fig. 1b). Each compartment is associated with an overrepresented biological process or cell type, with **W** recording the gene weights measuring the relevance of each gene for each compartment and **H** recording the compartment weights measuring the relevance of each compartment for each sample. To derive stable **W** and **H** for optimal compartments, and circumvent the need to heuristically determine $K$, DECODER was developed as a sophisticated framework which integrates multiple runs of a carefully designed NMF-based pipeline based on an increasing number of factors ($\tilde{K}$). In each run, a gene weight seed ($\mathbf{W}'$) is first trained by $R$ iterations of the NMF algorithm, followed by applying final NMF and nonnegative least squares (NNLS) projection (Fig. 1c, see the Methods section). With the deconvolved factors at multiple runs of increasing $\tilde{K}$, factor linkages are established by linking the most associated factors between adjacent runs. Finally, compartments are determined dynamically by evaluating factor scores and score patterns along each branch of linked factors, allowing compartments to be located at different runs of $\tilde{K}$ (Fig. 1d, see the Methods section). For this study, factors are used to refer to the direct output from each run at $\tilde{K}$, while compartments are used to refer to the final identified biological components from all the factors.

In a PDAC microarray data set containing primary tumor, metastatic and normal samples, DECODER identified 14 major compartments as a blinded determined solution, which accurately reproduced each of the compartments deconvolved previously using NMF at $K = 14$ based on empirical trialing[16] (Fig. 1e). For the matched compartments in the current and previous result, the gene weights show good correlation (median: 0.847, range: 0.761–0.924), and the top 250 genes show a large overlap (median: 158, range: 85–207) (Supplementary Fig. 1a, b). Similar to previous results, enrichment of deconvolved compartment weights for patient samples exhibit excellent agreement with known tissue labels[16] (Supplementary Fig. 1c). Notably, metastatic samples show enriched weights for compartments of both the metastatic and primary sites, e.g., liver metastases show enriched DECODER weights for the liver compartment, as well as PDAC basal tumor and classical tumor compartments. This is in

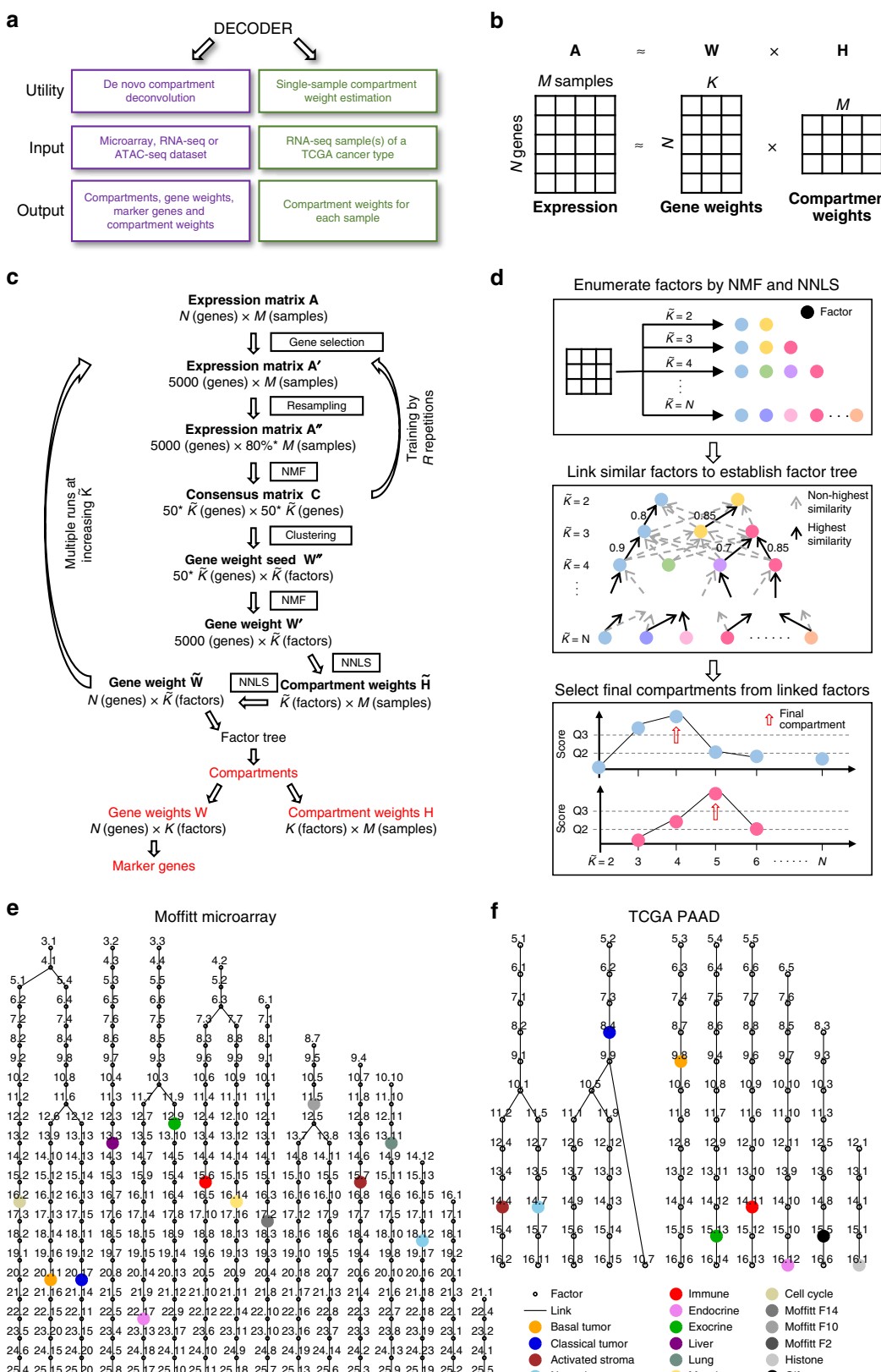

**Fig. 1** Overview of DECODER and the application of it. **a** The utility of DECODER includes de novo compartment deconvolution for a data set, and single-sample compartment weight estimation for TCGA cancers. **b** De novo deconvolution of DECODER aims to factorize an expression matrix **A** into two matrices of **W** (gene weights for compartments or gene weights) and **H** (compartment weights for samples or compartment weights). **c** The main steps for de novo deconvolution of DECODER. **d** The identification of final compartments from factors in multiple runs of $\tilde{K}$. **e**, **f** The factor trees for the Moffitt microarray data set and the TCGA PAAD data set, respectively. The resultant major compartments are colored, with the dropped factors, minor and unstable compartments denoted by black circles. Factor ID is used for labeling

agreement with other studies, where liver metastases were found to be molecularly conserved compared with their primary tumors in PDAC[21,22]. The high level of agreement is reassuring and suggests that DECODER may be used to enable the automated identification of compartments instead of involving labor-intensive manual annotation and empirical determination of $K$ at multiple separate runs.

**Deconvolution of 33 TCGA cancer types**. We then applied DECODER to each of the TCGA RNA-seq data sets of 33 cancer types for de novo compartment deconvolution. The compartments identified within cancer type were pooled, resulting in 269 compartments in total, with a median of 9 compartments in each cancer type (range: 4–14) (Supplementary Data 1, complete results available on GitHub: https://github.com/laurapeng/decoder/results). Then the ranked list of genes for each compartment was subjected to gene set analysis using the Molecular Signatures Database v3.1 (MSigDB)[23] for the biological interpretation and annotation of compartments. We manually annotated five compartments common across cancer types by using MSigDB: immune, basal tumor, activated stroma, histone, and olfactory. Analysis of overlap in the top 250 genes of 269 compartments showed four clusters of closely correlated compartments across cancer types: immune, activated stroma, histone, and basal tumor (Supplementary Fig. 2a, b).

For the TCGA pancreatic adenocarcinoma (PAAD) data set, nine major compartments were identified using DECODER de novo deconvolution. We analyzed each compartment by examining enriched MSigDB gene sets, and found that seven of the nine were dominant compartments similarly defined in the microarray data set for PDAC: basal tumor, classical tumor, activated stroma, normal stroma, immune, endocrine, and exocrine (Fig. 1f). For each compartment, genes with distinctly large weights were selected as marker genes (Supplementary Fig. 3a, Supplementary Data 2, see the Methods section). To investigate the representativeness of the marker genes, we used a linear model described by DSA[15] to calculate the fractions of seven dominant compartments using these marker genes. We found that the leukocyte fraction and ESTIMATE immune score were correlated with the estimated immune compartment fraction (Supplementary Fig. 3b, c). In addition, both of the ABSOLUTE and methylation-based tumor purity were highly correlated with the sum of the basal tumor and classical tumor compartment fractions (Supplementary Fig. 3d, e), and the ESTIMATE stromal score was highly correlated with the sum of the activated stroma and normal stroma fractions (Supplementary Fig. 3f). These findings demonstrate that DECODER can automatically and robustly determine biological compartments in a given data set de novo, and identify representative marker genes for each compartment.

**Compartment weights (H) derived from de novo deconvolution**. A matrix of compartment weights for samples (**H**) was also derived from the de novo deconvolution (Supplementary Data 3). We hypothesized that for each sample, a larger weight is associated with a larger representation for a specific compartment. Indeed, in TCGA PAAD, when we evaluated the hematoxylin and eosin (H&E) slides for the presence or absence of immune infiltrate and tertiary lymphoid structures within the tumor, samples with high immune weights showed apparent infiltration, which was absent in low immune-weighted samples (Fig. 2a). Quantitatively, samples containing tertiary lymphoid structures ($n = 37$) showed significantly higher immune weights than those with no tertiary lymphoid structures ($n = 84$) (Fig. 2b). We also found that normalized DECODER weight for the immune

compartment was highly correlated with leukocyte fraction (Fig. 2c) and ESTIMATE immune score (Fig. 2d). In addition, high immune weights predict better overall survival in TCGA PAAD, as well as a subset of the 33 cancer types[24] (Supplementary Fig. 4). For tumor compartments, we demonstrated high correlation between the sum of basal tumor and classical tumor weights, and tumor purity based on both ABSOLUTE (Fig. 2e) and methylation (Fig. 2f). Similarly, the ESTIMATE stromal score is mirrored by the sum of activated stroma and normal stroma weights (Fig. 2g).

Based on previous tumor subtype calls (the Moffitt schema) derived by consensus clustering using exemplar genes for these samples[19], we show that Moffitt Basal-like samples ($n = 37$) are associated with higher DECODER basal compartment weights and Moffitt Classical samples ($n = 113$) with higher DECODER classical compartment weights (Fig. 2h, i). Hereafter, for clarity, basal and classical tumor (lowercase) refer to DECODER-derived compartments, and Basal-like and Classical (uppercase) refer to the categorical subtypes. Similarly, higher ratios or differences of basal versus classical tumor compartment weights are associated with Moffitt Basal-like subtypes (Fig. 2j, k). To determine the clinical significance of basal versus classical tumor compartment weights, we compared the utility of compartment weights as a clinical variable. We found that the ratio (B./C.) ($p = 0.049$, Cox proportional-hazards model) but not difference (B.-C.) ($p = 0.073$, Cox proportional-hazards model) between DECODER basal and classical tumor weight is associated with survival as continuous variables.

To determine the accuracy of using DECODER compartment weights to perform binary classification of tumor subtypes, we calculated the area under the receiver operating curve (AUC) using Moffitt tumor subtype calls as the gold standard. We found that basal compartment weight alone, B./C., and B.-C., have similarly high area under the receiver-operating curve (AUC 0.94–0.96, Fig. 2l). Because B./C. shows the second highest AUC after B.-C. and is associated with outcome as a continuous variable, we then determined a threshold for the ratio (B./C. = 1) to reach optimal accuracy to call Moffitt subtypes (Fig. 2m), and optimal significance to differentiate patient outcome (Fig. 2n). Interestingly, the subtype-specific survival differences between the Basal-like ($n = 28$) and Classical samples ($n = 122$) were even more pronounced for the calls derived by DECODER B./C. than by the consensus clustering based calls of Moffitt schema (Fig. 2o, p).

**Single-sample compartment weight estimation**. Since the compartment weights for TCGA data set were obtained through de novo deconvolution which requires a data set with multiple samples and relatively large amounts of computing time during the training process, we next developed a method to deconvolve RNA-seq samples and calculate the compartment weights without the need to apply the de novo deconvolution. This method was built using the deconvolved gene weights for compartments (**W**) and applying NNLS to indicate the compartment weights for even a single sample (see the Methods section). Ten-fold cross-validation demonstrated good reproducibility of compartment weights derived from gene weights compared with those derived from de novo deconvolution (Supplementary Fig. 5a, b). We applied this algorithm to calculate the compartment weights for the COMPASS trial[20] and ICGC PACA-AU RNA-seq data sets[18] based on gene weights derived in de novo deconvolution of the TCGA PAAD data set (Supplementary Data 3).

Interestingly, in COMPASS ($n = 50$), where samples were microdissected, the endocrine, exocrine, and immune weights were significantly lower than those in the TCGA PAAD and ICGC

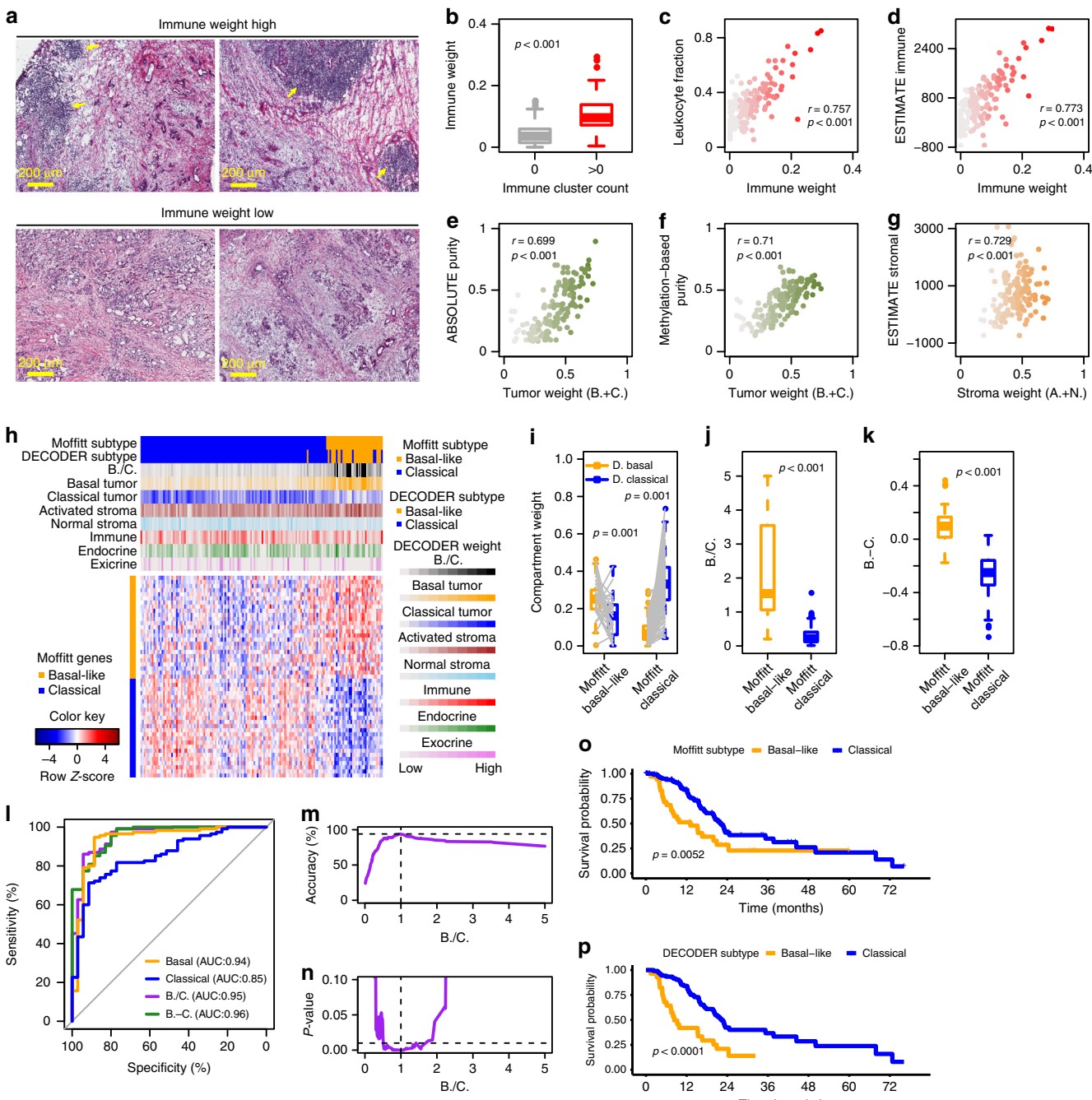

**Fig. 2** Deconvolved compartment weights for samples in TCGA PAAD data set. **a** Representative H&E-stained PDAC tumor sections showing immune cell infiltration (arrows) in samples with high immune vs low immune weight samples separated by the median. **b** Quantification of immune clusters by H&E staining was correlated with immune weight (two-sided Wilcoxon rank-sum test). **c, d** Correlations of immune weight with the leukocyte fraction (Pearson correlation) and ESTIMATE immune score (Spearman correlation). **e, f** Correlations of the sum of basal tumor and classical tumor weights, with tumor purity estimated by ABSOLUTE and methylation (Pearson correlation). **g** Correlation of the sum of activated stroma and normal stroma weights with ESTIMATE stromal score (Spearman correlation). **h** Associations of compartment weights with the tumor subtype calls. Heatmap shows consensus clustering of TCGA samples using 50 Moffitt tumor exemplar genes. Colored tracks show compartment weights, the ratio between basal tumor and classical tumor weights (B./C.), tumor subtypes called by B./C. (DECODER subtype), and tumor subtypes called by the clustering-based Moffitt schema (Moffitt subtype). **i** Basal tumor and classical tumor weights are compared in Moffitt Basal-like and Classical samples (paired two-sided Wilcoxon rank-sum test). **j, k** The ratio (B./C.) and difference (B.-C.) between basal tumor and classical tumor weights are compared in Moffitt Basal-like and Classical samples (two-sided Wilcoxon rank-sum test). **l** The receiver-operating curve (ROC) for basal tumor weight, classical tumor weight, B./C. and B.-C. in classifying subtypes. The area under the ROC (AUC) is shown for each parameter. **m** Threshold determination for B./C. to classify subtypes based on accuracy using Moffitt tumor subtype calls as gold standard. **n** Threshold determination for B./C. to classify subtypes based on significance to differentiate survival. **o, p** Kaplan–Meier plots of overall survival in patients with resected PDAC for Moffitt and DECODER tumor subtypes. Patients with B./C. >= 1 were categorized as Basal-like, while those with B./C. < 1 as Classical (log-rank test). Box plots in **b, i–k** show the median (center line) and interquartile range (box), with whiskers denoting 1.5 times the interquartile range above and below the upper and lower quartile, respectively

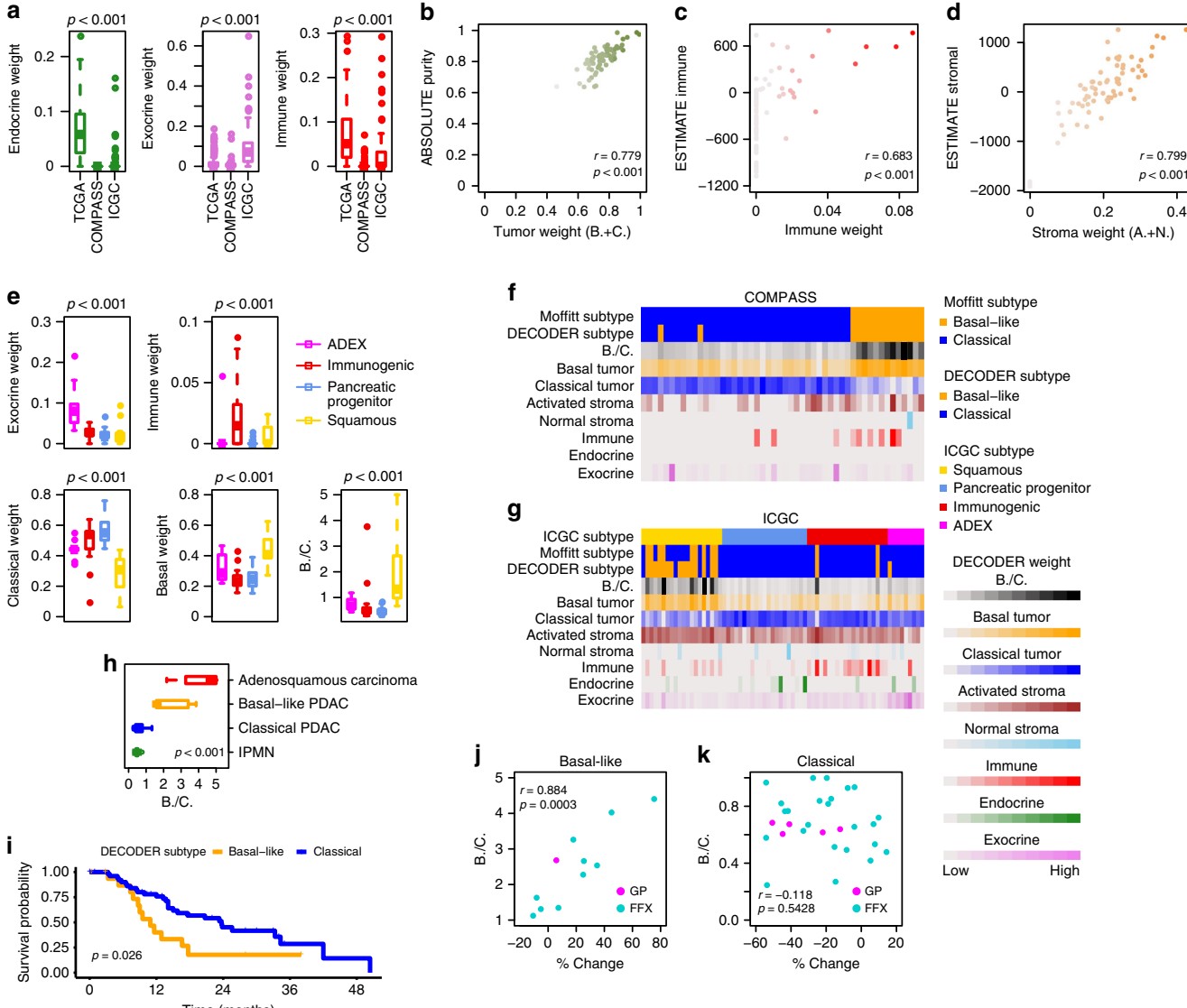

**Fig. 3** Single-sample compartment weight estimation in COMPASS and ICGC PACA-AU RNA-seq data set. **a** Endocrine, exocrine, and immune weights in the COMPASS (microdissected), ICGC (PACA-AU), and TCGA (PAAD) data sets (Kruskal–Wallis test). **b**–**d** Correlations of DECODER estimated tumor weight (the sum of basal tumor and classical tumor weights), immune weight, and stroma weight (the sum of activated stroma and normal stroma weights) with ABSOLUTE tumor purity (Pearson correlation), ESTIMATE immune score (Spearman correlation), and ESTIMATE stromal score (Spearman correlation), respectively. **e** Exocrine, immune, classical tumor, and basal tumor weights for the ICGC-subtyped ADEX, Immunogenic, Pancreatic Progenitor, and Squamous samples in the ICGC PACA-AU RNA-seq data set (Kruskal–Wallis test). **f**, **g** Associations of compartment weights with the tumor subtype calls. Colored tracks show compartment weights, the ratio between basal tumor and classical tumor weights (B./C.), tumor subtypes called by B./C. (DECODER subtype), tumor subtypes called by the clustering-based Moffitt schema (Moffitt subtype) and subtypes called by ICGC (ICGC subtype). For COMPASS, samples are ordered by consensus clustering using 50 Moffitt tumor exemplar genes. For ICGC, samples are ordered by ICGC subtypes. **h** B./C. for ICGC samples grouped by intraductal papillary mucinous neoplasm (IPMN), Classical PDAC, Basal-like PDAC and adenosquamous carcinoma (Kruskal–Wallis test). **i** Kaplan–Meier plot of DECODER subtypes in the ICGC data set (Log-rank test). Patients with B./C. > = 1 were categorized as Basal-like, while those with B./C. < 1 as Classical. **j**, **k** Correlations of the percent change (% change) in size of tumor target lesions from baseline, with B./C. in DECODER Basal-like and Classical samples (Pearson correlation). One Basal-like sample with an unstable DNA subtype was removed. Treatment of gemcitabine plus nab-paclitaxel (GP) and modified-FOLFIRINOX (FFX) was colored, respectively. Box plots in **a**, **e**, **h** show the median (center line) and interquartile range (box), with whiskers denoting 1.5 times the interquartile range above and below the upper and lower quartile, respectively

data sets (Fig. 3a). In the ICGC data set ($n = 70$), similar to our TCGA PAAD results, we found that ABSOLUTE-based tumor purity, ESTIMATE-based immune and stromal scores correlate with respective compartment weights derived by DECODER (Fig. 3b–d). For the two acinar cell carcinoma samples in this data set, we found that the exocrine weights were 4.78-fold and 6.96-fold higher than the mean of the exocrine weights in all other samples. This suggests that the calculated DECODER weights can

accurately capture the biological composition of a sample. In addition, the ICGC-subtyped ADEX samples ($n = 9$) are associated with higher exocrine weights, and the immunogenic samples ($n = 20$) are associated with higher immune weights (Fig. 3e). These results may suggest that ADEX and Immunogenic samples are characterized by an overrepresentation of exocrine and immune cells in the respective samples, which may confound the identification of tumor-specific subtypes[19,25].

In both the COMPASS and ICGC data sets, the DECODER basal weight, classical weight, as well as B./C. were associated with the Moffitt subtypes (Fig. 3f, g; Supplementary Fig. 5c–f). As expected, invasive intraductal papillary mucinous neoplasm (IPMN, $n = 10$) showed the lowest B./C., while the adenosquamous carcinoma samples ($n = 4$) showed the highest, suggesting that the B./C. is able to identify the extremes of tumor histology[26] (Fig. 3h). In the ICGC data set, where survival data were available, the DECODER Basal-like ($n = 18$) and Classical samples ($n = 52$) subtyped by B./C. were differentially associated with patient outcome (Fig. 3i). In addition, B./C. was found to significantly correlate the percentage of tumor size change after treatment in DECODER Basal-like subtype samples in COMPASS, where patients were treated with either modified-FOLFIRINOX (FFX) or gemcitabine plus nab-paclitaxel (GP) (Fig. 3j). However, this trend was not observed in the DECODER Classical subtype samples (Fig. 3k). These results demonstrated that the DECODER compartment weights may faithfully and accurately recapitulate the differences in the biological make-up of a single sample, thus enabling the accurate prediction of clinical variables.

**Application on TCGA PanCan ATAC-seq data set.** Previous PanCan analysis on an ATAC-seq data set of 23 human cancers has identified 18 distinct clusters of samples, which showed strong concordance with the published multiomic iCluster scheme[27,28]. These studies found both homogeneous clusters for single-tumor types, and heterogeneous clusters formed by mixed-tumor types arising from the same organ systems or with similar features, with some cancer types split into multiple clusters. We therefore interrogated whether DECODER can deconvolve the PanCan ATAC-seq data set containing 759 replicates from 410 unique samples into compartments that reflect inner biological composition. DECODER identified 22 major compartments. Unsupervised clustering on normalized compartment weights revealed clusters of known labels of cancer types, ATAC-seq-based clusters and iClusters (Fig. 4a). For example, the compartment denoted as D28.19:LIHC was found to show enriched weights in liver hepatocellular carcinoma (LIHC) samples (Fig. 4a), as well as in the ATAC-seq based cluster A9:Liver and the iCluster C26:LIHC (Fig. 4a, c, d).

Ten compartments were found to show uniquely higher weights in single cancer types, namely LIHC (D28.19:LIHC), skin cutaneous melanoma (D29.22:SKCM), adrenocortical carcinoma (D28.15:ACC), pheochromocytoma and paraganglioma (D29.9:PCPG), prostate adenocarcinoma (D29.10:PRAD), thyroid carcinoma (D21.12:THCA), uterine corpus endometrial carcinoma (D28.6:UCEC), testicular germ cell tumors (D28.23:TGCT), lung adenocarcinoma (D29.27:LUAD), and bladder urothelial carcinoma (D29.24:BLCA) (Fig. 4b). Similarly, we found their compartment weights to be the highest for ten respective ATAC-seq clusters and iClusters (Fig. 4c, d). This suggests that DECODER identified cancer-specific compartments similar to previous findings which identified cancer-specific clusters using ATAC-seq-based clustering and iClusters[27,28].

DECODER also identified organ system-associated compartments. For instance, brain (D29.13:GBM + LGG[Brain]), pan-kidney (D14.8:KIRC + KIRP[Pan-Kidney]), pan-gastrointestinal (D21.5:COAD + STAD + ESCA[Pan-GI]), digestive (D29.26: STAD + ESCA[Digestive]), and pan-squamous (D16.12:Pan-Squamous). For both of the brain tumors (glioblastoma multiforme [GBM] and brain lower grade glioma [LGG]), D29.13: GBM + LGG(Brain) showed exclusively high weight. Intriguingly, the compartment of D29.9:PCPG also showed relatively high weight in GBM and LGG, reflecting the fact that they may be

anatomically similar. Comparing with ATAC-seq clusters and iClusters, D29.13:GBM + LGG(Brain) was associated with A5: Brain, and C11:LGG(IDH1 mut) or C23:GBM/LGG(IDH1 wt), respectively. D14.8:KIRC + KIRP(Pan-Kidney) was distinctly highly weighted for kidney clear cell carcinoma (KIRC) and kidney renal papillary cell carcinoma (KIRP), represented as a pan-kidney compartment, which is associated with A1:Kidney/ Bile duct and C28:Pan-Kidney. Similarly, D21.5:COAD + STAD + ESCA(Pan-GI) and D29.26:STAD + ESCA(Digestive) were found to be related to the pan-GI system and clusters, with D21.5:COAD + STAD + ESCA(Pan-GI) exhibiting the highest weights in colon adenocarcinoma (COAD) and stomach adenocarcinoma (STAD), and the second highest weights in esophageal carcinoma (ESCA). For ESCA, which may have squamous morphology components, the compartment with highest weight was annotated as the pan-squamous compartment (D16.12:Pan-Squamous), showing the strongest associations with the squamous clusters in ATAC-seq and iCluster as well. D16.12: Pan-Squamous is also the most highly represented in head and neck squamous cell carcinoma (HNSC), lung squamous cell carcinoma (LUSC), and cervical and endocervical cancers (CESC), known to have squamous histologies. Interestingly, while D16.12:Pan-Squamous showed the second highest weights in BLCA, we found that the cancer-specific D29.24:BLCA compartment was overrepresented as well. This is in agreement with the finding that BLCA has very diverse iCluster memberships[28].

Compartments D27.24:BRCA-Basal, D20.18:BRCA-Her2 +, D28.11:BRCA-Chr8qAmp, and D27.13:BRCA-Luminal were all found to be associated with breast invasive carcinoma (BRCA) ($N = 141$), enabled by sample sufficiency in the data set and as expected by its known heterogeneity[29]. Less prominent compartments were found in cholangiocarcinoma (CHOL, $n = 10$), TGCT ($n = 18$), and mesothelioma (MESO, $n = 13$), which may be due to the small samples sizes available.

## Discussion

DECODER is an integrated framework, which we developed to perform de novo compartment deconvolution for any data set with nonnegative values, and conduct efficient compartment weight estimation for even a single sample of cancer types in TCGA. Standard NMF methods pre-define the number of factors ($K$) and assume the presence of $K$ compartments in a data set. In contrast, the de novo compartment deconvolution of DECODER is fluid and allows each of the compartments in a data set to be identified at runs of different $\tilde{K}$, facilitating the identification of each compartment to be more robust. This obviates the need for prior knowledge of the compartments and the number of them in a data set, since compartments vary in different cancers or tissues and certain compartments present in one may be absent in another.

We applied DECODER to deconvolve each of the 33 cancer types in the TCGA RNA-seq data sets. This has resulted in the identification of cancer-type-specific marker genes, which may lead to more accurate estimation of certain compartment fractions. As far as we know, current methods for deconvolution of tumor, stroma, and immune fractions use the same set of curated marker genes for all cancer types[7,14]. Furthermore, cancer-specific marker genes may be studied as cancer-specific biomarkers. The deconvolved mRNA compartments and respective marker genes for 33 cancer types are readily accessible on GitHub [https://github.com/laurapeng/decoder/results].

Our detailed examination on the resultant compartments of TCGA PAAD data set demonstrated the robust identification of biological compartments in PDAC, as defined by previously

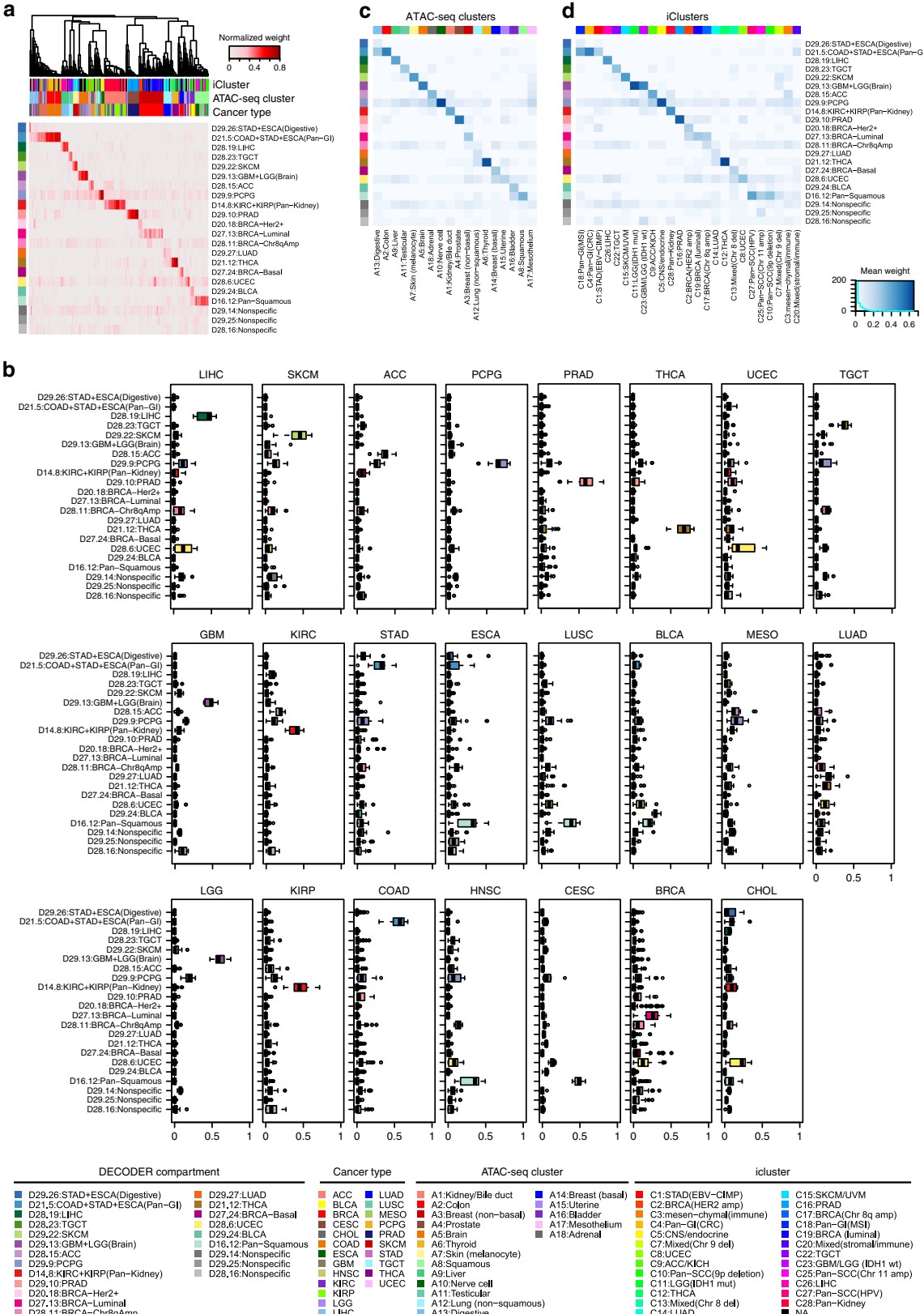

known knowledge[16]. We also showed that DECODER compartment weights for tumor, stroma, and immune were highly correlated with respective measurements by previous independent methods based on copy-number variations, methylation, and expression. In addition to the previously described compartments of basal-like tumor, classical tumor, activated stroma, normal

stroma, immune, endocrine, and exocrine, we also identified two new compartments, which we annotated as olfactory and histone. Interestingly, a study has associated the olfactory transduction pathway with pancreatic cancer risk[30], and additional studies have found overexpression of olfactory receptors in multiple cancer types, including prostate[31,32], bladder[33], and breast[34]

**Fig. 4** De novo deconvolution on TCGA ATAC-seq PanCan data set. **a** Unsupervised hierarchical clustering on samples (columns) using compartment weights on the heatmap. The darker the color, the higher the weight for the respective compartment and sample. Tracks above show cancer types, ATAC-seq-based cluster calls, and iCluster calls for the samples. Compartments (rows on the heat map) were manually ordered so that the clusters of enriched weights are shown on the diagonal. The same order of the compartments is maintained in **b**–**d**. The name of each compartment is generated by concatenating an uppercase letter "D", factor ID, a colon (:) and the manual annotation for the compartment. **b** Compartment weights for each cancer type. Box plots show the median (center line) and interquartile range (box), with whiskers denoting 1.5 times the interquartile range above and below the upper and lower quartile, respectively. **c**, **d** Associations of samples weights (rows) with ATAC-seq-based clusters and iClusters (columns). In each ATAC-seq-based cluster or iCluster, the means of the compartment weights are shown on the heatmap. Columns on the heat maps were manually ordered so that the best match with DECODER compartments are shown on the diagonal

cancers. In agreement with these studies, we found that our olfactory compartment was indeed present in prostate adeno-carcinoma (PRAD), bladder urothelial carcinoma (BLCA), as well as several other cancer types (Supplementary Data 1). Similarly, the histone compartment was identified in multiple cancer types. Further investigation will be required to determine if the olfactory and histone compartments are true biological compartments or artifacts of inherent noise in RNA-seq.

Unlike regular clustering-based methods, DECODER provides the possibility of examining a sample multidimensionally via the compartment weights, instead of forcing a given sample to a specific cluster. This provides more detailed information regarding the biological composition of a sample, which can be used for comparison across samples and data sets. In PDAC, absolute consensus for transcriptomic subtypes by different taxonomies has not been achieved, with the exocrine-like/ADEX and immunogenic subtypes at the center of the controversies[26]. By applying DECODER to the COMPASS and ICGC data sets, we demonstrated that in COMPASS, where samples were microdissected, there is comparatively less of the exocrine and immune compartments. In ICGC, the exocrine compartment weights correlate with the exocrine-like/ADEX subtype while immune compartment weights correlate with the Immunogenic subtype. These findings agree with the association of low-purity samples with exocrine-like/ADEX and Immunogenic subtypes[19] and suggest that the exocrine-like/ADEX and immunogenic subtypes may be explained by the presence of the nontumor compartments. In addition, in the more heterogenous ICGC data set, basal and classical tumor weights correlate well with IPMN versus adenoquamous carcinomas. Therefore, DECODER may facilitate the better elucidation of the underlying biology of molecular subtypes.

Furthermore, relying on the results from the initial de novo deconvolution in TCGA PAAD, compartment weight estimation in ICGC and COMPASS was single-sample based, and therefore feasible in the clinical setting. Similarly, for any of the 33 cancer types in TCGA, DECODER can be used to estimate the compartment weights for a new given sample without the need to perform de novo deconvolution. Thus, DECODER is a powerful tool to break down a new tumor sample with known origin.

DECODER may be applied to any solid or hematologic malignancy, as well as liquid biopsies, and may also be used for platforms other than gene expression. From a computational perspective, DECODER could be applied to any data type with nonnegative values derived from multiple platforms, e.g. ChIP-seq data, copy-number variations, DNA methylation data and single-cell sequencing platforms. As a proof of concept, we applied DECODER on the PanCan ATAC-seq data set containing 23 cancer types in a combined fashion, and identified compartments associated with cell-of-origin and organ systems, which highly reproduced previous clustering-based interpretations of the PanCan analysis[27,28]. Similar to our application of DECODER in RNA-seq, deconvolution of the PanCan ATAC-seq data set can also be applied to a single new sample. One limitation of our

PanCan analysis may be the unequal number of samples in different cancer types (range of number of samples: 7–141), which may lead to an unbalanced number of compartments identified for different cancers. Therefore, it is possible that more unbiased compartment information may be derived in the setting of a greater number of samples, especially for the cancers with smaller sample sizes.

From a clinical perspective, the ability of DECODER to identify sample compartments without the a priori assumption of an actual number of compartments, makes it a powerful tool for comparing tumor heterogeneity across samples and for the very challenging clinical conundrum of cancers of unknown primary. Cancers of unknown primary are metastatic cancers where the primary site of origin cannot be identified. Optimal treatment of these cancers remains a challenge, as knowledge of the primary site guides treatment decisions. We and others have previously shown that metastatic samples are molecularly conserved compared with their primary tumors[16,22]. Our successful deconvolution on PanCan ATAC-seq data supports the fact that profiled open-chromatin regions are extremely tissue specific[35], making ATAC-seq a promising method for distinguishing a complex mixture of cancer types. More ATAC-seq cohorts in metastatic samples will be needed to further validate our results. The application of DECODER using ATAC-seq data may provide more biological information that can better guide treatment for cancers of unknown primary and will need to be studied in the context of future clinical trials.

## Methods
**NMF seed training**. To handle the stochastic nature of the NMF, a stable gene weight seed ($\mathbf{W}''$) was trained before the application of a final NMF (Fig. 1c). $\mathbf{W}''$ is a $50 * \tilde{K} \times \tilde{K}$ matrix of $50 * \tilde{K}$ rows of genes and $\tilde{K}$ columns of factors. For de novo deconvolution of microarray and RNA-seq expression profiles recorded as $\mathbf{A}$ (a $N \times M$ matrix of $N$ genes and $M$ samples), highly expressed genes in the top third quartile were subjected to selection of the 5000 most variable ones resulting in $\mathbf{A}'$ (a $5000 \times M$ matrix). $R$ ($R = 10,000$ by default and in this study) repetitions of fivefold resampling (~80% of the samples, $\mathbf{A}''$) were performed. For each of the $R$ data partitions, unsupervised NMF was executed with 20 randomly initialized instances of NMF using the multiplicative update NMF solver for ten steps using the built-in NMF function in MATLAB (R2017b). The pair of gene weights and compartment weights with the lowest residual solution from these 20 instances were then used to seed NMF of $\mathbf{A}''$ to convergence with the alternating least-squares solver. The result contains a gene weight matrix for the current number ($\tilde{K}$) of factors (genes as rows and factors as columns). Based on this gene weight matrix, for each factor (column), the genes were ranked in descending order of the weight difference between the current factor weight and the largest weight in the rest of the factors. The top 50 genes for any factor were then recorded in a gene-by-gene consensus matrix $\mathbf{C}$ ($50 * \tilde{K} \times 50 * \tilde{K}$) by approximation due to possible duplicated top genes in multiple factors. This consensus matrix represents the frequency of the genes to be determined as the top genes, and was then used for hierarchical clustering to yield $\tilde{K}$ gene clusters. These $\tilde{K}$ gene clusters were used to create a seed matrix $\mathbf{W}''$, so that the top genes for the respective cluster have loading value of 1, with the rest of the genes having loading value of 0.01. This gene weights seed $\mathbf{W}''$ was then used to seed a final NMF, with a robust deconvolution of gene weights ($\mathbf{W}'$) and compartment weights ($\mathbf{H}'$). Of note, none of $\mathbf{A}'$, $\mathbf{A}''$, $\mathbf{W}'$, $\mathbf{W}''$, $\mathbf{H}'$ or $\mathbf{H}''$ involves matrix transpose for $\mathbf{A}$, $\mathbf{W}$ or $\mathbf{H}$; instead, they represent related but different matrices. Similarly, 8000 most highly expressed and variable ATAC-seq peaks were involved in the seed training process.

**Compartment weights and gene weights projection**. The final NMF outputs two matrices, i.e., gene weight matrix $\mathbf{W}'$ (a $5000 * \tilde{K}$ matrix of 5000 rows of genes and $\tilde{K}$ columns of factors), and compartment weight matrix $\mathbf{H}'$ (a $\tilde{K} \times M$ matrix of $\tilde{K}$ rows of factors and $M$ columns of samples). Subsequently, NNLS was used to find

$$\arg\min_{\mathbf{h}_i} ||\mathbf{W}'\tilde{\mathbf{h}}_i - \mathbf{a}'_i||_2 \text{ subject to } \tilde{\mathbf{h}}_i \geq 0, \tag{1}$$

where $\tilde{\mathbf{h}}_i$ is the $i$th column/sample in $\tilde{\mathbf{H}}$ (a $\tilde{K} \times M$ matrix) to be determined, $\mathbf{a}'_i$ is the $i$th column/sample in $\mathbf{A}'$ (a $5000 \times M$ matrix). Of note, while discarding $\mathbf{H}'$, $\tilde{\mathbf{H}}$ is considered to record the final compartment weights for each factor at the current number of factors ($\tilde{K}$) from the de novo deconvolution.

With $\tilde{\mathbf{H}}$ derived by estimating each $\tilde{\mathbf{h}}_i$, similarly, NNLS was used to find

$$\arg\min_{\mathbf{w}_j} ||\tilde{\mathbf{H}}\tilde{\mathbf{w}}_j - \mathbf{a}_j||_2 \text{ subject to } \tilde{\mathbf{W}}_j \geq 0, \tag{2}$$

where $\tilde{\mathbf{w}}_j$ is the $j$th row/gene in $\tilde{\mathbf{W}}$ (a $N \times \tilde{K}$ matrix) to be determined, and $\mathbf{a}_j$ is the $j$th row/gene in $\mathbf{A}$ (a $N \times M$ matrix). As a result, $\tilde{\mathbf{W}}$ is considered to record the final gene weights for each factor at the current number of factors ($\tilde{K}$) from the de novo deconvolution. This gene weight matrix ($\tilde{\mathbf{W}}$) is further used for ranking of genes, calculation of factor scores, and annotation of compartments.

**Minimum and maximum of $\tilde{K}$ to be considered**. The range of increasing $\tilde{K}$ can be customized by the user. By default, the step of factor deconvolution starts with $\tilde{K} = 2$ and ends when the median of factor scores are <0.5 for more than three consecutive runs. In addition, starting from $\tilde{K} = 2$, the run at $\tilde{K}$ will begin to be considered until the median of the factor scores at $\tilde{K}$ are >0.5.

**Factor score calculation and linkage establishment**. The factor ID is generated by concatenating the number of factors at current run ($\tilde{K}$), a dot (.), and an unduplicated number from 1 to $\tilde{K}$ randomly assigned by the algorithm. To score each factor, genes for each factor at each run of $\tilde{K}$ are ranked to determine the top genes. Specifically, for a factor ($i$th column in $\tilde{\mathbf{W}}$) at each run of $\tilde{K}$, the genes were ranked in descending order of the weight differences between the loading value in the $i$th column and the largest loading value in the rest of the columns. Based on the ranking, the top 250 genes are identified. The overlapping percentages converted to decimals of the top 250 genes between each of the factors at $\tilde{K}$ and each of the factors at $\tilde{K} - 1$ (similarity) were calculated. For the $i$th factor at $\tilde{K}$, if the $ii$th factor at $\tilde{K} - 1$ showed the largest overlap (highest similarity) and was greater than 0.1, then the overlapping percentage converted to decimal (similarity) is determined as the score for the $i$th factor at $\tilde{K}$. If the two factors $i$th at $\tilde{K}$ and $ii$th at $\tilde{K} - 1$ were considered to be associated (i.e., largest overlapping percentage and greater than 0.1), then a link is established between them (Fig. 1d). However, if the similarity between the $i$th factor at $\tilde{K}$ and the factors at $\tilde{K} - 1$ is <0.1, the $i$th factor at $\tilde{K}$ will be considered a newly emerged factor. If there is no association between the $i$th factor at $\tilde{K}$, and any of the factors at $\tilde{K} + 1$, then no linkage is established as the respective factor was considered too split or diluted to be detected in the resolution of $\tilde{K} + 1$. In this way, linkages of associated factors in each run of a different $\tilde{K}$ are established.

**Compartments identification from factors**. The linking of all the related factors at multiple runs of $\tilde{K}$ leads to the placement of all the identified factors on a tree-like plot (Fig.1e, f). Along each linkage, the pattern of the factor scores is examined to determine the optimal factor along each linkage. We assume that the consistently relative high scores of adjacently linked factors indicate the existence of a stable biological component. Therefore, compartment identification was performed, so that a resultant compartment: (a) is on a linkage with >2 factors on it, (b) shows a score greater than the third quartile ($Q_3$) of all the scores, (c) has more than two continuous (adjacently linked) (major) or one (unstable) factor(s) greater than the median quartile ($Q_2$) of all the scores, (d) is with the maximum score among the continuous factors with score higher than $Q_2$ (candidates in a high-score factor block), and (e) has the largest number of factors in the high-score block or has the greatest score if multiple high-score blocks and candidates found. Finally, if a major compartment is found to have over 100 overlapping genes in the top 250 with another major compartment, and both of them have the same linkage, the compartment identified at larger $\tilde{K}$ is then labeled as a "minor" compartment. For each of the final compartments, respective columns of gene weights are extracted from $\tilde{\mathbf{W}}$ of the respective run at $K$, and combined to form a final gene weights matrix $\mathbf{W}$. $\mathbf{W}$ is saved as a reference for later single-sample compartment weight estimation. Similarly, rows of $\tilde{\mathbf{H}}$ corresponding to the resultant compartments are extracted to from the final compartment weights $\mathbf{H}$ for the samples in the bulk measurement $\mathbf{A}$.

**Gene analysis and compartment annotation**. For each compartment, the marker genes were identified as the top genes with positive normalized weight differences exponentially greater than the rest, which are above the geometric inflection point on the ranking curve (Supplementary Fig. 3a). Ranked genes in each compartment were subjected to gene set analysis using annotated gene sets from MSigDB v3.1[21],

assessed by the Kolmogorov–Smirnov statistic with Benjamini–Hochberg correction for significance[16]. For RNA-seq data, final compartments of TCGA PAAD were annotated based on MSigDB terms and prior knowledge. For the remaining cancer types, the immune, activated stroma, basal tumor, olfactory, and histone compartments were annotated according to MSigDB, with the remaining compartments unannotated (Supplementary Data 1).

**Single-sample compartment weight estimation**. Given a new bulk measurement $\mathbf{B}$, which is a $N \times M$ matrix of $N$ rows of genes and $M$ columns of samples, and pre-computed gene weights $\mathbf{W}$ derived from de novo deconvolution, which is a $N \times K$ matrix of $N$ rows of genes and $K$ columns of compartments, NNLS is used to find

$$\arg\min_{\tilde{\mathbf{h}}_i} ||\mathbf{W}\mathbf{h}_i - \mathbf{b}_i||_2 \text{ subject to } \mathbf{h}_i \geq 0, \tag{3}$$

where $\mathbf{h}_i$ is the $i$th column/sample in $\mathbf{H}$ to be determined, $\mathbf{b}_i$ is the $i$th column/sample in $\mathbf{B}$.

**Compartment weight normalization**. For pancreatic cancer, weights for seven compartments (basal tumor, classical tumor, activated stroma, normal stroma, immune, endocrine, and exocrine) were normalized so that the sum of them equals to 1 for each sample. For PanCan ATAC-seq data set, weights for the 22 major compartments were normalized so that the sum of them equals to 1 for each sample.

**Ten-fold cross-validation**. In TCGA PAAD, samples were randomly partitioned into ten subsets. De novo deconvolution was then performed on each combination of ninefold of samples to derive final gene weights. Then for each onefold of samples, the compartment weights were then estimated by the gene weights derived by the other ninefold of samples by the nonnegative least square (NNLS) algorithm. Finally, the estimated compartment weights were compared with those derived from performing de novo deconvolution on the whole data set.

**Statistical information**. Two-sided paired Wilcoxon rank-sum test was used for comparison of basal and classical tumor weights in subtyped samples. Two-sided Wilcoxon rank-sum test was used to compare ratios and differences of basal and classical tumor weights in subtyped samples. The Kruskal–Wallis test by ranks was used for testing whether the compartment weights originated from the same distribution when more than two categories of samples were involved. In correlation analysis, Spearman correlation was used to compare DECODER-derived weights with ESTIMATE-derived scores, where the scales were largely different. Otherwise, Pearson correlation was involved. For survival analysis, continuous variables were analyzed by Cox proportional-hazards regression model, and categorical variables were analyzed by log-rank test.

**Reporting summary**. Further information on research design is available in the Nature Research Reporting Summary linked to this article.

## Data availability

A published collection of microarray data (GSE71729 and GSE21501) from a previous study[16] was used for comparison between de novo deconvolution of DECODER and previous NMF run in the study. TCGA normalized RNA-seq gene expression data from 33 cancer types were downloaded from the Broad Institute FIREHOSE portal [http://gdac.broadinstitute.org]. For the PAAD data set, out of 183 samples deposited in the portal, we included 150 white-listed PDAC samples for further analysis[19]. For the rest of the cancer types, all of the downloaded samples were used for the de novo deconvolution analysis. The normalized ATAC-seq counts within the pan-cancer peak set were downloaded from the Genomic Data Commons website [https://gdc.cancer.gov/about-data/publications/ATACseq-AWG]. Loci with mean counts across samples greater than the overall mean, and standard deviation (SD) of counts across samples greater than the mean of all SD ($n = 105,138$) were subjected to de novo deconvolution of DECODER. Seventy samples with matched mRNA subtype calls from Bailey et al.[18], and RNA-seq data, as well as clinical data (PACA-AU) were downloaded from ICGC data portal [http://dcc.icgc.org/]. RNA-seq data of 50 PDAC samples that underwent laser capture microdissection in COMPASS trial[20] were obtained from European Genome-Phenome Archive (EGA, EGAS00001002543) with granted access. A reporting summary for this article is available in Supplementary Table 1.

## Code availability

A Matlab repository for DECODER is available at GitHub [https://github.com/laurapeng/decoder], with the utilities of de novo deconvolution and single sample compartment weight estimation. MATLAB R2017b is recommended for use. All the downloaded data sets are also deposited in this repository, along with all the deconvolution results in this study. The configure files for de novo deconvolution (Moffitt microarray, TCGA RNA-seq, and TCGA PanCan ATAC-seq data sets), and single-sample compartment weight estimation (COMPASS PDAC and ICGC pancreatic cancer data sets) are provided as Supplementary Tables 2–6. In addition, an R package for single sample compartment weight estimation for TCGA cancers is readily accessible at GitHub [https://github.com/laurapeng/decoder]. R version 3.4 is recommended for use.

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

## Acknowledgements

We would like to thank the University of North Carolina at Chapel Hill and the Research Computing group for providing computational resources and support that have contributed to these research results. This work was supported by grant U24-CA211000 and R01-CA199064 (X.L.P. and J.J.Y.). We would like to thank Dr. Naim Rashid (Department of Biostatistics, University of North Carolina at Chapel Hill) for inspiring discussions. We would like to thank Mr. Chong Jin (Department of Biostatistics, University of North Carolina at Chapel Hill) for his tremendous help in providing mathematical insight during the paper revision.

## Author contributions

J.J.Y. and X.L.P. designed the study. X.L.P. and R.A.M. performed the computational analyses. R.J.T. and K.E.V. performed the histology analyses. J.J.Y. and X.L.P. wrote the paper. All authors critically reviewed and commented on the paper.

## Competing interests

J.J.Y., X.L.P., R.A.M., R.J.T., and K.E.V. are inventors of DECODER.
