## [Peer Review File · Nature Communications]

Reviewers' comments:

Reviewer #1 (Remarks to the Author):

The authors applied non-negative matrix factorization (NMF) algorithm to deconvolve heterogeneous tissue samples using gene expression data. One of the challenges of using NMF is that the number of factors (K) needs to be specified a priori. For a large dataset, testing different K's could be computationally costly. In this paper, the authors further extended the NMF method by developing a streamlined framework that automatically determines the optimal K.

Major comments

The authors refer to the factors from the NMF analysis as "compartment" which has a specific meaning in biology. The broader readership might find that term confusing. For the 33 TCGA Pan-cancer datasets, the authors identified 269 cancer-specific "compartments". Are those all the major factors or only the cancer-specific ones? In deconvolution of tissue samples, one typical goal is to estimate cell subsets in the bulk tissue. Can the authors make clear what biological concept(s) they believe the compartments correspond to?

What are the "cancer type-specific derived compartments"? Do you mean factors with large gene weights in specific cancer types? It would be nice to provide a graphic illustration of these weights – e.g., how different tumor types cluster using those gene weights. Although knowing different tumor types were associated with different "compartments" is useful, it might be more interesting to know the relative proportions of the same "compartments" in different tumor types and how the differences are related to clinical outcomes, for example.

Throughout the manuscript, much of the discussion on clustering of the weights is not descriptive (e.g., D28.19:LIHC) and is hard for the readers to connect it to the context.

I assume that "gene weight" and "sample weight" are the outputs of the NMF analysis. Both terms need to be defined mathematically as well as the rationale for their biological interpretations.

What is the "factor lineage"? is it a scalar? Please define it mathematically.

In line 109, what are "factor score" and "factor patterns"? Are you referred to the score matrix in factor analysis? It would be useful to define the terms mathematically.

In line 392, it is not clear what "run of 10000 iterations" was referring to. To NMF? If so, in what settings? The built-in NMF in MATLAB has a few parameters to choose from. Details on how the NMF was run would be helpful.

In lines 392-393, what does "frequency of the genes" mean? How did you obtain the seed matrix W' exactly?

In line 401, I don't understand why the authors tried to minimize $Ax-y$? When 'x' is the gene-specific weight for a given sample, to reconstruct 'y' (bulk gene expression for the sample), one should minimize $Ax-y$, where A represents the sample-specific weights. But the authors define A as the original data matrix. I am sure the authors did it correctly. Any clarification would be helpful.

The authors stated that DECODER can work with single samples. I am not sure if I understand exactly how it would work. Given a bulk measurement B (a vector), you would run NMF to get the W' , and then NNLS on B to get the sample weight? It would be helpful if the procedure was clearly described.

How do the authors know the step-wise minimization reached the global minimum and not a local minimum?

Minor comments

In line 132, What is "FARMER_BREAST_CANCER_CLUSTER_5"?

What does "classical tumors" mean?

Reviewer #2 (Remarks to the Author):

The authors have developed an algorithm for deconvolution of tumour expression data (or related functional data such as open chromatin data) in additive components using nonnegative matrix factorisation (NMF). This has been done previously by the authors and others - the new feature here is that the algorithm has a automatic criterion to select the number of compartments K by detecting when there is not enough evidence in the data to increase K to $K+1$.

This is a useful contribution to the field. Previous applications of NMF have not focussed on the robust identification of a specific number K of factors. The proposed algorithms looks like a good approach and will greatly simplify the application of NMF, reducing the amount of human expert interaction and subjective choice required to apply such a method.

The method is demonstrated on large-scale datasets from microarrays, RNA-Seq and ATAC-Seq experiments and the components obtained from one dataset are shown to be useful when applied to a different dataset. There are some novel compartments identified and validation against other independent measures of tumour composition give strong evidence that the identified components are meaningful.

My only concern is that the code is written in the proprietary language matlab which limits users to those with matlab licenses and unfortunately this limits the ability to include the full tool within standard open source workflows, although the pre-trained method to apply to individual samples is available in R at least. Nevertheless, despite this negative aspect overall I think this is still a nice contribution.

Reviewer #3 (Remarks to the Author):

The manuscript by Peng and co-authors presents a new computational method that deconvolutes the compartments of tumors. Using available transcriptional data, a systematic methodology, and cross-validation the authors demonstrate that their method can identify compartments in pancreatic adenocarcinoma PDAC first. They then demonstrate that the method is transferable to the 33 cancer types of the TCGA RNA-seq datasets. The analyses and the results are presented in a clear and transparent manner.

Points that should be addressed to enhance impact relate to the applicability of the method and therefore its relevance:

The authors suggest that their work is relevant because it "may have implications for identifying the tumor of origin for cancers of unknown primary". Two aspects should be considered and demonstrated to support this.

- 1) The primary tumor may or may not have a different compartment structure compared to metastases. The authors should demonstrate that a biopsies from metastases can be sufficiently reliably deconvoluted, show robust compartments, and can, therefore, be assigned to a primary.
- 2) Related, but with extended relevance, RNAseq data of biopsies from carcinomas of unknown primary should be obtained de novo and used to identify the primary tumor site. This is relevant,

because the ultimate extension of the work would be to assign cancer specific treatment to patients with cancers that would have previously been treated with the regimen for unknown primaries and to investigate if this improves outcome.

Minor points.

3) Line 245 Fig. 3b is referenced rather than Fig. 4b.

4) Line 273 typographical error "agreement"

Reviewer #1 (Remarks to the Author):

The authors applied non-negative matrix factorization (NMF) algorithm to deconvolve heterogeneous tissue samples using gene expression data. One of the challenges of using NMF is that the number of factors (K) needs to be specified a priori. For a large dataset, testing different K's could be computationally costly. In this paper, the authors further extended the NMF method by developing a streamlined framework that automatically determines the optimal K.

We thank the reviewer for his/her thorough review.

Major comments

The authors refer to the factors from the NMF analysis as “compartment” which has a specific meaning in biology. The broader readership might find that term confusing. For the 33 TCGA Pan-cancer datasets, the authors identified 269 cancer-specific “compartments”. Are those all the major factors or only the cancer-specific ones? In deconvolution of tissue samples, one typical goal is to estimate cell subsets in the bulk tissue. Can the authors make clear what biological concept(s) they believe the compartments correspond to?

Thank you for this comment. NMF factorizes a non-negative matrix into factors to achieve dimension reduction. In the biology context, the original matrix (e.g. each TCGA cancer RNA-seq dataset) measures the expression of over 20,000 genes in hundreds of samples. The dimensionality reduction leads to the identification of compartments which are associated with stable biological processes or cell types, characterized by clusters of over-expressed genes. Of note, compartments do not have to be mutually exclusive in a single sample, and the fractions of them in each sample indicate the sample composition. To further clarify this point, we have added more detail at lines 97-100.

We apologize for making the definition of “cancer-specific” compartments confusing here. The “269 cancer-specific compartments” are specific in the context that they are identified in parallel by applying *de novo* deconvolution of DECODER to 33 TCGA tumor RNA-seq datasets separately 33 times. Of note, major compartments in each cancer types were identified separately, after which they were simply pooled to form the 269 compartments in total. We assume that the reason causing confusion here is the fact that some compartments have similar annotations in different cancer types, e.g. immune, histone, basal tumor and activated stroma. Although these compartments seem to be not specific to a certain cancer, we did not merge or combine them in the 269 compartments. For example, the pancreatic cancer immune compartment is independent of the liver cancer immune compartment. This is because, although some compartments share similar biological functions, they do not exhibit exactly the same gene signatures. We have revised the description in the manuscript at lines 131-135.

What are the “cancer type-specific derived compartments”? Do you mean factors with large gene weights in specific cancer types? It would be nice to provide a graphic illustration of these weights – e.g., how different tumor types cluster using those gene weights. Although knowing different tumor types were associated with different “compartments” is useful, it might be more interesting to know the relative proportions of the same “compartments” in different tumor types and how the differences are related to clinical outcomes, for example.

Thank you for this comment. Similar to the answer to the last point, “cancer type-specific derived compartments” are just compartments identified in individual applications of the *de novo* deconvolution in 33 cancer types separately, instead of considering their specificity across cancer types. To “provide a graphic illustration” of “how different tumor types cluster using those gene weights”, we analyzed the percentage of overlaps in the top 250 genes between each pair of the 269 compartments. The overlapping percentages were subjected to unsupervised hierarchical clustering shown on the heat map (Supplementary Fig. 2a), revealing 4 clusters of closely resembled compartments across cancer types, i.e. immune, histone, activated stroma and basal tumor. For these closely resembled compartments, the top genes were only partially overlapped (Supplementary Fig. 2b). Nonetheless, the majority of the compartments are divergent across cancer types, raising the importance of tumor type specific deconvolution.

We wholeheartedly agree that it would be extremely interesting to study the relative proportions of the same compartments in different tumor types. However, the estimation of the relative proportion of a compartment relies on the accurate annotation of all the major biological compartments as the background. This can be accomplished similar to what we show using TCGA PAAD, where we identified 9 major compartments, but determined 7 out of them to be the dominantly contributing biological compartments, manually excluding the histone and olfactory compartments. Therefore, this annotation will require more in-depth knowledge of each

specific cancer type, which we consider to be out of the scope for this manuscript and our current expertise. Nevertheless, it would be very feasible for other groups to further study and annotate the cancer types of interest, using the provided GSEA annotation as guidance, as well as marker genes for each compartment for all cancer types.

To dissect the clinical relevance of the same compartments in different cancer types, we specifically studied the immune compartment across cancer types. We classified the patients into immune-high and immune-low group separating by the median of immune weight in each cancer type respectively. For different cancer types, the immune compartment shows variability in predicting prognosis, the interpretation of which requires in-depth knowledge for each individual cancer type (Supplementary Fig. 4).

We have incorporated the results in the manuscript at lines 140-142 and 171-173.

Throughout the manuscript, much of the discussion on clustering of the weights is not descriptive (e.g., D28.19:LIHC) and is hard for the readers to connect it to the context.

We thank the reviewer for this comment. We used compartment weights for clustering in Fig. 4a. In Fig. 4a, unsupervised clustering on normalized compartment weights revealed clusters of known labels of cancer types, ATAC-seq-based clusters and iClusters (Fig. 4a). For example, the compartment denoted as D28.19:LIHC was found to show enriched weights in liver hepatocellular carcinoma (LIHC) samples (Fig. 4a), as well as in the ATAC-seq based cluster A9:Liver and the iCluster C26:LIHC (Fig. 4a,c,d). The description has been revised at lines 251-255.

I assume that “gene weight” and “sample weight” are the outputs of the NMF analysis. Both terms need to be defined mathematically as well as the rationale for their biological interpretations.

Thank you for this comment. We have revised the term of “sample weight” to “compartment weight” whenever it’s mentioned. To better define “gene weight” and “sample weight” (now “compartment weight”) biologically, we have revised the manuscript at lines 97-100 to clarify that each compartment is associated with an over-represented biological process or cell type, with **W** recording the gene weights measuring the relevance of each gene for each compartment and **H** recording the compartment weights measuring the relevance of each compartment for each sample.

What is the “factor lineage”? is it a scalar? Please define it mathematically.

Thank you for this comment. We assume the reviewer is asking about “factor linkage” that we proposed. “factor linkage” is neither a scalar nor a vector. However, to clarify, “factor linkage” can be interpreted as similar to a vector in a way that, the \tilde{K} at which the factor is derived could be regarded as the direction, and the factor score could be regarded as the magnitude. Factor linkages are established by linking similar factors in different runs of de novo deconvolution. The definition of factor linkage and the way to establish it can be found in lines 468-475 following the definition of factor score.

In line 109, what are “factor score” and “factor patterns”? Are you referred to the score matrix in factor analysis? It would be useful to define the terms mathematically.

Thank you for this comment. “Factor score” is different from the score matrix. “Factor score” is defined as the maximal percentages of overlapped genes of current factor at \tilde{K} with factors at $\tilde{K} - 1$ converted to decimal. This score measures whether the current factor out of \tilde{K} factors is associated with any of the factors in the previous run of de novo deconvolution at $\tilde{K} - 1$ and how similar they are. This definition is at lines 459-467 with enhanced descriptions added. “factor patterns” is the pattern of factor scores in each linkage. We enhanced the description of it, which can be found in lines 478-488.

In line 392, it is not clear what “run of 10000 iterations” was referring to. To NMF? If so, in what settings? The built-in NMF in MATLAB has a few parameters to choose from. Details on how the NMF was run would be helpful. In lines 392-393, what does “frequency of the genes” mean? How did you obtain the seed matrix **W** exactly?

Thank you for these comments. To clarify “run of 10000 iterations”, “frequency of the genes” “how to obtain the seed matrix W ” and MATLAB NMF parameters, we re-organized the section of “**NMF seed training**” in the manuscript accordingly at lines 408-432.

The code for the NMF parameters can be found on GitHub

(<https://github.com/laurapeng/decoder/tree/master/utills>). Specifically, the code to perform the NMF in the training iterations are in `nmmf_unmix_quick.m` as follows.

```
opt = statset('MaxIter', 10, 'Display', 'final');  
[genesigs, samplesigs] = nmmf(Data, factors, 'options', opt, 'replicates', 20, 'algorithm', 'mult');  
opt = statset('MaxIter', 1000, 'Display', 'final');  
[genesigs, samplesigs] = nmmf(Data, factors, 'options', opt, 'w0', genesigs, 'h0', samplesigs);
```

In line 401, I don't understand why the authors tried to minimize $Ax-y$? When 'x' is the gene-specific weight for a given sample, to reconstruct 'y' (bulk gene expression for the sample), one should minimize $Ax-y$, where A represents the sample-specific weights. But the authors define A as the original data matrix. I am sure the authors did it correctly. Any clarification would be helpful.

We thank the reviewer for pointing this out. We realized that the descriptions on how we used NNLS to infer gene and sample weights in both *de novo* deconvolution and single-sample weights estimation are not clear enough. Moreover, we made an ambiguous definition for A when we define A as the bulk measurement input and use it again in $Ax-y$. Therefore, we re-organized the respective section of “**Compartment weights and gene weights projection**” (lines 435-450) to better reflect the methodology and rationale involved.

The authors stated that DECODER can work with single samples. I am not sure if I understand exactly how it would work. Given a bulk measurement B (a vector), you would run NMF to get the W , and then NNLS on B to get the sample weight? It would be helpful if the procedure was clearly described.

Thank you for this comment. For a given single new bulk measurement B , only NNLS is needed to get the sample weights (now relabeled as compartment weights for this revision). This is because gene weights (W) was already derived from a certain cancer type using prior deconvolution. NNLS will then be used to solve $\min ||Wx-y||$ to derive x . To clarify this procedure, we added a separate section of “**Single-sample compartment weight estimation**” to better illustrate how we derive single-sample based weights at lines 508-515.

How do the authors know the step-wise minimization reached the global minimum and not a local minimum?

Thank you for this comment. We assume the reviewer is referring to the difficulties in solving NMF. Unfortunately, NMF is generally NP-hard¹. Therefore, most methods will optimize alternatively over one of the two factors, W or H , while keeping the other fixed. While this subproblem is convex and relatively easy to solve, only the convergence to stationary points may be guaranteed, instead of the global minimum¹, and the solution can be highly dependent on the initial values. Nevertheless, NMF is still one of the best choices to deconvolve the underlying components without any additional information. In our method, to circumvent the instability of NMF solutions, we trained a robust NMF seed of W , which guaranteed the convergence of the final NMF output so that the final marker gene lists are stable.

Minor comments

In line 132, What is “FARMER_BREAST_CANCER_CLUSTER_5”?

It is an annotation term from Molecular Signatures Database (MSigDB) that we found to be associated with the compartment of activated stroma in multiple cancer types. To make this less confounding, we have removed the exact MSigDB terms at lines 138-139, since these annotation terms are available in the complete results at <https://github.com/laurapeng/decoder/results>.

What does “classical tumors” mean?

In pancreatic ductal adenocarcinoma, the two tumor subtypes are denoted as “Basal-like” and “Classical” subtypes by identifying two tumor compartments, which were referred to as the compartment of “basal tumor” and “classical tumor” in the manuscript. We have added a word “tumor” after “basal” when we talk about “basal and classical tumor” at line 174, if this is where the confusion is caused.

Reviewer #2 (Remarks to the Author):

The authors have developed an algorithm for deconvolution of tumour expression data (or related functional data such as open chromatin data) in additive components using nonnegative matrix factorisation (NMF). This has been done previously by the authors and others - the new feature here is that the algorithm has a automatic criterion to select the number of compartments K by detecting when there is not enough evidence in the data to increase K to $K+1$.

This is a useful contribution to the field. Previous applications of NMF have not focussed on the robust identification of a specific number K of factors. The proposed algorithms looks like a good approach and will greatly simplify the application of NMF, reducing the amount of human expert interaction and subjective choice required to apply such a method.

The method is demonstrated on large-scale datasets from microarrays, RNA-Seq and ATAC-Seq experiments and the components obtained from one dataset are shown to be useful when applied to a different dataset. There are some novel compartments identified and validation against other independent measures of tumour composition give strong evidence that the identified components are meaningful.

We thank the reviewer for the good comments. And we appreciate the recognition toward the novelty of our method and the broad application of our method to multiple platforms and datasets.

My only concern is that the code is written in the proprietary language matlab which limits users to those with matlab licenses and unfortunately this limits the ability to include the full tool within standard open source workflows, although the pre-trained method to apply to individual samples is available in R at least. Nevertheless, despite this negative aspect overall I think this is still a nice contribution.

We agree with the reviewer that the fact that the de novo deconvolution utility of DECODER is only written in Matlab will limit its usage, and we apologize for that. We have spent a great deal of effort in trying to re-write DECODER in R. However, to train robust NMF gene weight seed, 10,000 iterations of simple NMF runs were involved, which is extremely resource- and time-consuming and makes it impossible to derive a final result from R. R is not good at dealing with large amount of loops. because of this, DECODER was written in Matlab so it can run smoothly. For users who are not familiar with Matlab, the coding examples for running DECODER is provided in the supplementary information.

Reviewer #3 (Remarks to the Author):

The manuscript by Peng and co-authors presents a new computational method that deconvolutes the compartments of tumors. Using available transcriptional data, a systematic methodology, and cross-validation the authors demonstrate that their method can identify compartments in pancreatic adenocarcinoma PDAC first. They then demonstrate that the method is transferable to the 33 cancer types of the TCGA RNA-seq datasets. The analyses and the results are presented in a clear and transparent manner.

We thank the reviewer for his/her careful efforts in reviewing this manuscript.

Points that should be addressed to enhance impact relate to the applicability of the method and therefore its relevance:

The authors suggest that their work is relevant because it “may have implications for identifying the tumor of origin for cancers of unknown primary”. Two aspects should be considered and demonstrated to support this.

We agree it is pivotal to further demonstrate that DECODER is able to be used to identify the tumor of origin for cancers of unknown primary. To strengthen this argument, we have added further analysis. The specific concerns raised were addressed in details below.

1) The primary tumor may or may not have a different compartment structure compared to metastases. The authors should demonstrate that a biopsies from metastases can be sufficiently reliably deconvoluted, show robust compartments, and can, therefore, be assigned to a primary.

Thank you for this comment. The microarray dataset (under accession GSE71729 at Gene Expression Omnibus [GEO]) analyzed in Moffitt et.al. and in this study is a collection of samples from primary tumors,

metastatic tumors and normal tissues, e.g. metastatic tumors in liver, lung, spleen or lymph nodes². Both previous results and DECODER demonstrated the successful deconvolution of compartments associated with metastasis loci, i.e. compartments of liver, lung, muscle and immune (Supplementary Figure 2 in Moffitt et. al. and Supplementary Figure S1c in this manuscript). The respective factor/compartment weights for samples showed excellent agreement with known tissue. Notably, metastatic samples show enriched weights for compartments of both metastatic site and primary site, e.g. liver metastatic samples show enriched weights for the liver compartment, as well as PDAC basal tumor compartment and classical tumor compartment. This is in line with the study by Connor et al. using a different dataset, where liver metastases were found to be molecularly conserved compared to their primary tumor^{3,4}. To better illustrate this point in the manuscript, we have added Supplementary Fig. 1c and text at lines 119-126.

2) Related, but with extended relevance, RNAseq data of biopsies from carcinomas of unknown primary should be obtained de novo and used to identify the primary tumor site. This is relevant, because the ultimate extension of the work would be to assign cancer specific treatment to patients with cancers that would have previously been treated with the regimen for unknown primaries and to investigate if this improves outcome.

Thank you for this comment. We envision DECODER as being able to indicate the tumor-of-origin for metastatic samples based on two facts. First, metastatic tumors show gene programs for both tissue of metastatic sites and primary tumor site as illustrated in the reply to the reviewer #3's first question. Therefore, this task is biologically feasible. Second, the PanCan ATACseq dataset (23 cancer types) was deconvolved into distinct and distinguishable compartments associated with cancer type or organ system. This suggests that it is technically feasible. For a new tumor sample with unknown primary, the ATACseq data derived from this sample could be used to determine the compartment weights, with the over-represented compartments indicating the primary site. However, we are unable to locate publically available ATAC-seq data for metastatic tumor samples as a validation cohort. We are initiating a prospective collection of metastatic tumors for ATAC-seq but this will take a few years and is out of the scope of the current study.

We propose that PanCan ATAC-seq instead of PanCan RNA-seq will be needed to perform the identification of primary tumor site as open chromatin regions characterized by ATAC-seq are much more cell- and tissue-specific than gene expressions measured by RNA-seq. For example, MYC gene is activated in all of pancreatic cancer, T-ALL and colorectal cancer, but the enhancer regions (open chromatin regions) associated with this gene are all different in these cancers⁵. We did analyze PanCan RNA-seq in a combined fashion, but as expected, it did not reveal compartments as specific as using ATAC-seq (data not shown). These results were anticipated given the similarities in compartments across the individual runs of the 33 TCGA RNA-seq datasets (Suppl. Fig. 2). This pitfall is likely caused by the inherent nonspecific nature of gene expression, instead of the DECODER framework. With the wider application of ATAC-seq and availability of metastatic samples, will allow us to validate our results. We have revised our discussion to clarify these thoughts at lines 376-383 in the manuscript.

Minor points.

3) Line 245 Fig. 3b is referenced rather than Fig. 4b.

Thank you for pointing this out, we have correct the reference at line 262.

4) Line 273 typographical error "agreement"

Thank you for pointing this out, we have fixed the mistake at 291.

References

- 1 Gillis, N. in *arXiv e-prints* (2014).
- 2 Moffitt, R. A. *et al.* Virtual microdissection identifies distinct tumor- and stroma-specific subtypes of pancreatic ductal adenocarcinoma. *Nat Genet* **47**, 1168-1178, doi:10.1038/ng.3398 (2015).
- 3 Connor, A. A. *et al.* Integration of Genomic and Transcriptional Features in Pancreatic Cancer Reveals Increased Cell Cycle Progression in Metastases. *Cancer Cell* **35**, 267-282 e267, doi:10.1016/j.ccell.2018.12.010 (2019).
- 4 Martens, S. *et al.* Different shades of pancreatic ductal adenocarcinoma, different paths towards precision therapeutic applications. *Ann Oncol*, doi:10.1093/annonc/mdz181 (2019).
- 5 Sur, I. & Taipale, J. The role of enhancers in cancer. *Nat Rev Cancer* **16**, 483-493, doi:10.1038/nrc.2016.62 (2016).

REVIEWERS' COMMENTS:

Reviewer #1 (Remarks to the Author):

The authors articulated their choice of MATLAB over R in their implementation of DECODER. Unfortunately, the instructions on how to run their MATLAB code on Github is not very helpful.

It appears that the authors used the MATLAB built-in NMF function for their "multiplicative update NMF solver" instead of developing their own. If this was the case, it needs to be clarified.

Line 511, "N x K rows" should be "N rows".

Reviewer #2 (Remarks to the Author):

The authors have addressed my comments in the revised version.

Reviewer #3 (Remarks to the Author):

Peng and co-authors have improved their manuscript during revisions and have responded adequately to the points raised in my review.

REVIEWERS' COMMENTS:

Reviewer #1 (Remarks to the Author):

The authors articulated their choice of MATLAB over R in their implementation of DECODER. Unfortunately, the instructions on how to run their MATLAB code on Github is not very helpful.

We thank the reviewer for bringing up this concern. We have provided a README file in our GitHub repository (<https://github.com/laurapeng/decoder>) to show instructions on how to run the MATLAB scripts. Of note, the MATLAB script of `nmmf_unmix_quick.m` mentioned in our initial reply is a function that will be automatically called by the main script of DECODER for *de novo* deconvolution. For the user, only a simple line of code will be needed in the Matlab console, which is `Decon_de_novo(configFile)`. The demo file of `configFile` can be found both on GitHub (`config_de_novo.tsv`) and in the Supplementary Tables 2-6.

It appears that the authors used the MATLAB built-in NMF function for their “multiplicative update NMF solver” instead of developing their own. If this was the case, it needs to be clarified.

We thank the reviewer for this comment. We have now clarified that we used built-in multiplicative update NMF solver in the Methods section (line 447-448).

Line 511, “N x K rows” should be “N rows”.

We thank the reviewer for catching this. We have revised it accordingly.

Reviewer #2 (Remarks to the Author):

The authors have addressed my comments in the revised version.

We thank the reviewer for previous comments and for accepting our revisions.

Reviewer #3 (Remarks to the Author):

Peng and co-authors have improved their manuscript during revisions and have responded adequately to the points raised in my review.

We thank the reviewer for previous comments and for accepting our revisions.